# *Emiliania huxleyi*—Bacteria Interactions under Increasing CO_2_ Concentrations

**DOI:** 10.3390/microorganisms10122461

**Published:** 2022-12-13

**Authors:** Joana Barcelos e Ramos, Susana Chaves Ribeiro, Kai George Schulz, Francisco José Riso Da Costa Coelho, Vanessa Oliveira, Angela Cunha, Newton Carlos Marcial Gomes, Colin Brownlee, Uta Passow, Eduardo Brito de Azevedo

**Affiliations:** 1Group of Climate, Meteorology and Global Change, IITAA, University of the Azores, Rua Capitão d’Ávila, São Pedro, 9700-042 Angra do Heroísmo, Portugal; 2Centre for Coastal Biogeochemistry, School of Environmental Science and Management, Southern Cross University, P.O. Box 157, Lismore, NSW 2480, Australia; 3CESAM—Centre for Environmental and Marine Studies, Department of Biology, University of Aveiro, 3810-193 Aveiro, Portugal; 4The Marine Biological Association of the United Kingdom, The Laboratory Citadel Hill, Plymouth PL1 2PB, UK; 5Ocean Science, Faculty of Sciences, Memorial University of Newfoundland, St. John’s, NL A1C 5S7, Canada

**Keywords:** *Emiliania huxleyi*, CO_2_, coccolithophores, *Idiomarina abyssalis*, *Brachybacterium* sp., phytoplankton–bacteria interactions, changing ocean, functional profiling of marine bacteria

## Abstract

The interactions established between marine microbes, namely phytoplankton–bacteria, are key to the balance of organic matter export to depth and recycling in the surface ocean. Still, their role in the response of phytoplankton to rising CO_2_ concentrations is poorly understood. Here, we show that the response of the cosmopolitan *Emiliania huxleyi* (*E. huxleyi*) to increasing CO_2_ is affected by the coexistence with bacteria. Specifically, decreased growth rate of *E. huxleyi* at enhanced CO_2_ concentrations was amplified in the bloom phase (potentially also related to nutrient concentrations) and with the coexistence with *Idiomarina abyssalis* (*I. abyssalis*) and *Brachybacterium* sp. In addition, enhanced CO_2_ concentrations also affected *E. huxleyi*’s cellular content estimates, increasing organic and decreasing inorganic carbon, in the presence of *I. abyssalis*, but not *Brachybacterium* sp. At the same time, the bacterial isolates only survived in coexistence with *E. huxleyi*, but exclusively *I. abyssalis* at present CO_2_ concentrations. Bacterial species or group-specific responses to the projected CO_2_ rise, together with the concomitant effect on *E. huxleyi*, might impact the balance between the microbial loop and the export of organic matter, with consequences for atmospheric carbon dioxide.

## 1. Introduction

Earth’s climate and biosphere are strongly interlinked. The interactions established in the upper ocean between eukaryotic phytoplankton, bacteria and viruses play an important role in the pelagic energy flow and nutrient cycling [1,2] with consequences for biogeochemical cycles and feedback to climate.

The interactions established between marine phytoplankton and bacteria vary in complexity, from simply sharing the same environment, competing for the same resources and tightly relying on each other (mutualism) to living from the other in a host–parasite relationship. Phytoplankton growth and production rates affect organic matter characteristics, influencing bacteria community composition, abundance and production rates [3]. In parallel, bacteria community composition affects production rates and potentially functional redundancy and plasticity to changing environmental conditions [4]. As a result, relative abundances of some bacteria families correlate with phytoplankton biomass indicators during bloom events, while others remain unaltered [3]. CO_2_ concentrations are projected to reach about 700 µatm by the year 2100 according to the ‘intermediate scenario, RCP 6.0’ [5], with a concomitant decrease in seawater pH. This has been seen to affect the important players of marine elemental cycling, phytoplankton and bacteria, differently. In spite of the urgency of increasing knowledge on how these planktonic communities respond to global change, little is known about specific phytoplankton–bacteria interactions.

Coccolithophores, and particularly the cosmopolitan *Emiliania huxleyi* (*E. huxleyi*), have been shown to be sensitive to increasing CO_2_ concentrations (e.g., see review by [6,7]). How their response is affected by interactions with bacteria is poorly understood. Moreover, the response of the heterotrophic bacteria to increasing CO_2_ concentrations is also not well-known (e.g., [8]). Organic matter degradation can be significantly affected by changes in seawater pH due to changes in the efficiency of hydrolytic enzymes [9]. However, ocean acidification experiments often consider bacteria within entire planktonic communities in mesocosms (e.g., [10]) or shipboard incubations [11], complicating the disentanglement of specific bacterial responses and interactions. Relatively recent studies addressed bacteria associated with phytoplankton cultures, demonstrating the establishment of a core microbiome [12] in which the *Roseobacter* clade is often well represented [13] and that the interactions vary with environmental conditions, time and partner species. Examples of this are the known frequent associations of Marinobacter and Marivita [14], the studied symbiotic/commensal relationships [13,15] and the *E. huxleyi* relationships with the symbiotic and pathogenic α-proteobacteria *Phaeobacter inhibens* and *Phaeobacter gallaeciensis* (*Roseobacter clade*) (references in [16]). However, there is still scarce information about single bacterial isolates or groups and virtually none considering the bacteria-specific phytoplankton interactions under increasing CO_2_ concentrations.

This study aims to address the role of interactions established between a coccolithophore and two bacteria with unknown responses to increasing CO_2_ concentrations. For this, we focused on *E. huxleyi* and two bacteria strains previously isolated from offshore Terceira island: (1) the γ-proteobacteria *Idiomarina abyssalis* (*I. abyssalis*), found in association with phytoplankton and sinking particles and with higher average relative abundance in the epipelagic [17]; and (2) the Actinobacterium *Brachybacterium* sp., potentially associated with planktonic communities and in the wake of sinking particles [18]. It is known that *E. huxleyi* produces higher concentrations of carbohydrates under stressful conditions, such as enhanced CO_2_ concentrations [19]. This could lead to an increase in the relative number of heterotrophic bacteria in the proximities of the phytoplankton cells. Still, it remains unclear how these interactions could be differently affected and affect increasing CO_2_ concentrations. Here, we hypothesised that: (1) the response of *E. huxleyi* to increasing CO_2_ concentrations is affected by coexisting bacteria; (2) the response of the chosen bacteria to increasing CO_2_ concentrations is affected by coexisting *E. huxleyi*; (3) the response of *E. huxleyi* to increasing CO_2_ concentrations is bacteria species-specific.

## 2. Methods

### 2.1. Experimental Setup

Monospecific cultures of the cosmopolitan coccolithophore *E. huxleyi* (371, CCMP, axenic) were grown semi-continuously until the experiment for a minimum of 10 generations, under two CO_2_ concentrations (average *p*CO_2_: ~475 and 1056 µatm, pH_total scale_ of ~7.99 and 7.69, respectively; Appendix A). Cultures previous to and during the experiment were grown in 0.1 µm sterile filtered North Atlantic seawater (salinity of 36) enriched with phosphate and nitrate, reaching 4.7 µmol L^−1^ and 83 µmol L^−1^, respectively (Appendix A), and with trace metals and vitamins following f/8 [20], at 20 °C, a photon flux density of 185 (+/− 10) µmol m^−2^ s^−1^ (supplied from OSRAM L 18W/840, Lumilux, coolwhite) and a 14/10 h light/dark cycle. The absence of bacteria in the *E. huxleyi* cultures, as well as contamination in the cultures with bacteria, was confirmed every ~4 days in the pre-cultures and during the experiments by means of microscopy and agar plating. Considering that *Brachybacterium* sp. forms yellow colonies and *I. abyssalis* transparent, it was easier to check agar plates for contamination throughout the experiment and in all treatments.

The two bacteria (*Idiomarina abyssalis* PhyBa_CO2_1 and *Brachybacterium* sp. PhyBa_CO2_2), chosen from our culture collection, were originally isolated by direct plating in Marine Agar (Difco, Le Pont de Claix, France). The isolates were analysed for their potential for carbohydrate fermentation (Appendix A) to assure functional diversity and then selected for the experiment. Specifically, *I. abyssalis* was isolated in April 2018 from an *E. huxleyi* strain isolated from surface water off Biscoitos (Terceira island, 38°80′05″ N, 27°25′93″ W) and belongs to the Proteobacteria phylum. Related strains were *Idiomarina abyssalis* strain KMM 227 and *Idiomarina loihiensis* strain L2TR (100% and 99% 16S rRNA sequence similarity, respectively). *Brachybacterium* sp. was isolated in May 2019 from surface samples off the coast of Terceira (38°38′00″ N 27°09′00″ W) and belongs to the Actinobacteria phylum. Its most closely related strains identified with 16S were *Brachybacterium paraconglomeratum* strain LMG 19861 and *Brachybacterium conglomeratum* strain J 1015 (both with 100% 16S rRNA sequence similarity). However, based on Average Nucleotide Identity (ANI) information, the description of a new species is being prepared.

The isolates were kept at −20 °C in 30% (*v/v*) glycerol and revitalised in Marine Broth prior to the experiments at 20 °C under the same light conditions as *E. huxleyi*. Bacteria cultures were reactivated in Marine Broth for ~12 generations (division cycles) before the start of the experiment. On the day of the experiment, bacteria strains were centrifuged for 10 min at 14,000 rcf, washed twice with sterile seawater and resuspended with the corresponding medium (present and future) to a concentration of approximately 10^7^ colony-forming units (CFU) mL^−1^ (determined with the Track-Dilution method) and left in the incubation chamber overnight at 20 °C, a photon flux density of 185 (+/− 10) µmol m^−2^ s^−1^ (supplied from OSRAM L 18W/840, Lumilux, coolwhite) and a 14/10 h light/dark cycle. The culture media were always acclimated to the temperature of the experiment before inoculation of the bacteria and *E. huxleyi*.

At the start of the experiment, the axenic *E. huxleyi* was inoculated first (final concentration 2300 to 5600 cells mL^−1^), followed by the intended bacteria (final concentration *I. abyssalis* ~2 × 10^5^ CFU ml^−1^ and *Brachybacterium* sp. ~9 × 10^4^ CFU mL^−1^). During the exponential phase, *E. huxleyi* cell abundances ranged from ~2 and 6 × 10^4^ cell mL^−1^ and *I. abyssalis* and *Brachybacterium* sp. were ~1 × 10^5^ CFU mL^−1^. The low cell concentrations minimised changes in seawater carbonate chemistry (average DIC drawdown of 2.6%). All cultures were manually vertically rotated (15 times, gently) one hour after the beginning of the light phase to avoid aggregation, sedimentation and self-shading. Sampling occurred at the beginning and after 4 and 13/14 days of incubation. At the end of the experiment (bloom, day 13/14), *E. huxleyi* reached a maximum of ~1 × 10^5^ cell mL^−1^ while *I. abyssalis* reached ~7 × 10^6^ CFU mL^−1^ and *Brachybacterium* sp. ~3 × 10^2^ (present CO_2_) and ~9 × 10^6^ (high CO_2_) CFU mL^−1^.

### 2.2. Carbonate System

Total alkalinity was measured with an open cell potentiometric titration following Dickson et al. [21], using a Metrohm Titrino Plus 848 equipped with an 869 Compact Sample Changer and corrected with certified reference material supplied by A. Dickson (batch 128). The pH_t_ was determined via two methods: (1) through a glass electrode (WTW, pH 340i), which was calibrated with a TRIS seawater buffer, supplied by A. Dickson and; (2) colorimetrically by adding known amounts of m-cresol purple [22,23].

The carbonate system was manipulated by the addition of specific amounts of NaHCO_3_ and HCl in a closed system following [24]. All carbonate chemistry parameters were calculated from measured salinity, temperature, phosphate concentrations and pH and TA using CO2sys [25], with the equilibrium constants determined by Mehrbach et al. [26] as refitted by Dickson and Millero [27] (Appendix A).

### 2.3. Nutrients

Samples were taken for the determination of dissolved inorganic nutrients at the beginning and the exponential phase. They were filtered through a polyethersulfone (PES) 0.2 μm syringe filter and stored at −20 °C until analysis. Concentrations of nitrate and phosphate were measured following Hansen and Koroleff [28], using a Varian Cary 50 spectrophotometer.

### 2.4. Cell Numbers and Growth Rates

*E. huxleyi* abundances were determined with an inverted microscope (Nikon Eclipse TS 100) at 200× magnification from samples fixed with buffered Lugol (2% final concentrations) shortly after sample collection and counting on average ~1102 cells per sample +/− 88 SE. Bacteria were quantified after 48 h by counting viable colony counts (colony forming units, CFU) from agar plates after dilution steps, as well as total numbers of cells through flow cytometry on day 4 (exponential). More specifically, on day 4, it was important to complement the counts of viable cells (CFU) that grow and are responsible for bacterial rates with total bacteria (flow cytometry), essential to calculating cellular quotas. Hence, for the cellular quota data, flow cytometry was chosen. This strengthens the quality of the data and, therefore, assures comparison with previous works. Flow cytometry was conducted on frozen, glutaraldehyde (0.6% final concentration) preserved samples. The suspensions were stained with Syto BC in DMSO (Invitrogen) and incubated in the dark for 5 min at room temperature, beads were added at known concentration and reading duration was controlled to optimise the measurements. Samples were analysed with a FACSCalibur on the following gain settings: FL1 = 650; FL2 = 650; FL3 = 650; SS = 450 (event rate always below 1000 events per second).

Cell division rate (µ) was calculated as:µ = (ln Ce − ln Ci)/∆t(1)
where Ce and Ci refer to end and initial numbers of cells, respectively, and ∆t to the duration of the incubation period in days.

### 2.5. Particulate Organic Matter and Cellular Element Quotas

Samples for cellular particulate total carbon (TPC), organic carbon (POC) and nitrogen (PON) were gently filtered (200 mbar) through pre-combusted GF/F filters (6 h, 450 °C) and stored at −20 °C until analysis. TPC and PON samples were directly dried (4 h, 60 °C) while POC filters were firstly exposed to an acidified environment inside an exicator with 1 cm HCl (35%) for 2 h and then dried (4 h, 60 °C). All filters were then packed in tin boats and analysed in a gas chromatograph (EURO EA Elemental Analyser, EUROVECTOR equipped with a thermal conductivity detector and an element analyser) following Sharp [29]. Particulate inorganic carbon (PIC) was calculated by subtracting POC from TPC. Samples were taken during the exponential phase (fourth day of incubation) to allow comparison with previous studies.

In monocultures, cellular quotas for each species were calculated for day 4 (exponential) from POC, PON and PIC and the respective cell numbers of *E. huxleyi*, *Bacteriastrum* sp. and *I. abyssalis*. In the co-cultures, cellular POC, PON and PIC quota of each species was estimated by assuming no change as opposed to the mono-cultures in two ways:*E. huxleyi*_quotas_ = (C/N_total_ − ((A_b_ × Q_b_))/A_Ehuxleyi_(2)
*Bacteria*_quotas_ = (C/N_total_ − ((A_Ehux_ × Q_Ehux_))/A_b_(3)
where C/N_total_ corresponds to the total particulate matter in the filter (µg L^−1^) carbon (C) or nitrogen (N), from which the portion of the other organism in the co-culture is subtracted. This was achieved by multiplying their A (abundance) in the co-culture with the cellular quotas (Q) determined for the monoclonal cultures of the same CO_2_ concentration. The result was then divided by the abundance (A) using microscopy for *E. huxleyi*/flow cytometry in the bacteria) of the species intended to be estimated. These calculations assumed that the cell quota of one of the two organisms is not affected by co-existence. Comparing the unaffected with the potentially affected quota provides the full range of the potential response. It is worth noting that measured POC and PON can also include exudates such as transparent exopolymer particles (TEPs). The percentage (estimated from A_Ehux_ × Q_Ehux_) of particulate organic matter corresponding to *E. huxleyi* in the co-cultures was ~80% in all conditions, except in coexistence with *I. abyssalis* under enhanced CO_2_ concentrations (~90%). Therefore, changes in bacteria quotas would have less impact on *E. huxleyi* than the inverse (Appendix A).

Finally, bacterial cellular quotas of the mono-cultures were always calculated from untreated TPC filters since they do not have relevant particulate inorganic carbon. In the co-cultures, bacteria cellular quotas were determined from *E. huxleyi* total and organic carbon quotas.

### 2.6. Coccolith Morphology

Samples collected for scanning electron microscopy (SEM) were filtered through 0.45 µm nitrate cellulose membranes and dried in an exsiccator. Filters were glued to supports, coated with charcoal and processed with SEM. Resulting images were then analysed with image J. All photographs were taken at the same magnification and the measured lengths and widths were calibrated using the scale bar present in all pictures. An average of ~130 (with the exception of E + AU under high CO_2_ concentrations where only 6 coccoliths were measured) coccoliths were manually measured per condition, namely distal shield length (DSL) and width (DSW), and central area length (CAL) and width (CAW). From these measurements, the corresponding areas (distal shield area, DSA, and central area area, CAA) were calculated assuming an elliptical shape for the coccolith (e.g., [30]) as:DSA/CAA = π × ((length × width)/4)(4)

### 2.7. Extracellular Enzyme Activity

Extracellular enzyme activity was determined in triplicate in 96-well black plates for each bottle by means of 4-methyl-coumarinyl-7-amide (MCA) and 4-methylumelliferone (MUF) fluorogenic analogues using a high throughput plate reader approach (following [31,32]). The activities of protease, β-glucosidase, α-glucosidase, phosphatase, lipase and chitinase were measured using L-leucine-7-amino-4-methylcoumarin, 4-methylumbelliferyl β-D-glucopyranoside, 4-methylumbelliferyl α-D-glucopyranoside, 4-methylumbelliferyl phosphate, 4-methylumbelliferyl 4-oleate and 4-methylumbelliferyl N-acetyl-β-D-glucosaminide, respectively. Substrates were added to the samples at a 39 µM final concentration. Samples were taken at time 0 and after 3h of incubation under the experimental conditions (20 °C and 185 µmol m^−2^ s^−1^) and with minimum headspace by using a transparent, plastic, tight cover. After the incubation sample, fluorescence was measured at 365/460 nm (excitation/emission) for MUF and 380/440 nm (excitation/emission) for MCA with Fluostar Omega directly or after the addition of a ‘STOP’ buffer for the β- and α-glucosidase. Both final concentration and incubation duration were previously determined.

### 2.8. Utilisation of Carbohydrates

Fermentation of carbohydrates of the two bacterial isolates was determined using a colorimetric method prior to the experiment. Bacterial cells were suspended in culture medium to turbidity of 2 McFarland into a suspension medium with bromocresol purple as pH indicator. Thereafter, 100 µL of each sugar (Glucose, Frutose, Maltose, Trehalose, Ribose, Xylose, Dextrin, Starch, Lactose, Galactose, Arabinose, Rhamnose, Sucrose, Raffinose, Mannitol, Sorbitol and Inulin) was added at a concentration of 1% (*w/v*) to 400 µL of bacterial suspension. After incubation at 25 °C for 48 h, the color change was monitored. Results were expressed as clear positive (++), positive (+) or negative (−) according to color change (blue to yellow).

### 2.9. Bacteria Identification and Genome Analyses

#### 2.9.1. DNA Extraction and Sequencing

*Idiomarina abyssalis* PhyBa_CO2_1 and *Brachybacterium* sp. PhyBa_CO2_2 were cultivated in Marine Broth at 20 °C, in 2 consecutive cultures of 48 h each. After cultivation, genomic DNA for identification (Appendix A, and Appendix A) and sequencing was extracted by means of a commercial bacterial DNA isolation kit (PureLinkTM Microbiome, Carlsbad, CA, USA).

16S ribosomal RNA (rRNA) gene amplicons were generated with PCR using primers 27F (5′-AGAGTTTGGATCMTGGCTCAG-3′) and 1492R (5′-CGGTTACCTTGTTACGACTT-3′). The PCR reaction mixture contained 10 μL 5X GoTaq reaction buffer (Promega), 1 μL dNTPs (10 mM), 2.5 μL primer 27F (10 μM), 2.5 μL primer 1492R (10 μM), 0.5 μL GoTaq Polymerase (5 U/μL) (Promega) and 1 μL of the extracted DNA. Nuclease-free water (Promega) was added to reach a total reaction volume of 50 μL. The following conditions were used for the bacterial 16S rRNA gene amplification: initial denaturation at 98 °C for 10min followed by 35 cycles of denaturation at 98 °C for 20 s, annealing at 52 °C for 20 s, elongation at 72 °C for 45 s and a final extension step at 72 °C for 5 min. PCR products were purified using the Gene-JET PCR purification kit (Thermo Fisher Scientific, Waltham, MA, USA) and quantified using a Nanodrop 2000c spectrophotometer (Thermo Fisher Scientific, Waltham, MA, USA). The purified PCR products were sent for Sanger sequencing with primers 27F and 1492R (GATC Biotech, Cologne, Germany; now part of Eurofins Genomics Germany GmbH). Trimming (99% good bases, quality value > 20, 25-base window) and contig assembly were conducted with DNA Baser (version 3.5.4.2). Genome sequencing of strains was performed using the in Illumina Novaseq platform (150 bp paired-end reads). The genomes were sequenced at STAB VIDA (Lisbon, Portugal).

#### 2.9.2. Genome Assembly and Quality Control

The quality of the reads was assessed with FASTQC 0.11.5 (Andrews S. FASTQC: a quality control tool for high throughput sequence data; 2010; available online at: http://www.bioinformatics.babraham.ac.uk/projects/fastqc, accessed on 27 November 2022). Trimmomatic 0.38 was used to remove adapters and quality filtering with the parameters: Leading: 8; Trailing: 8; Slidingwindow: 4:15; and Minlen: 100 [33]. Genome sequences generated were de novo assembled with the SPAdes 3.15.2 [34]. The quality of the draft assemblies was evaluated using QUAST 5.0.2 [35]. Completeness and contamination of analysed genomes were estimated using CheckM 1.1.3 with the default set of marker genes [36]. Only genomes that were at least 95% complete and had no more than 5% contamination were used. Prediction of CDSs of the assembled genome was performed with the RAST 2.0 server using the ‘classic RAST’ algorithm [37].

#### 2.9.3. Phylogenetic and Phylogenomic Analyses

The phylogenetic analyses were conducted on sequences of the 16S rRNA genes of both strains and representatives of closely related described strains were retrieved from NCBI GenBank and included in the analysis. A phylogenetic tree was constructed using the web server Phylogeny.fr (http://www.phylogeny.fr/, accessed on 27 November 2022): sequences are aligned using Muscle, the alignment is curated using G-blocks and the phylogeny is established using PhyML-aLRT. Finally, the tree was drawn using TreeDyn [38].

For genome-wide assessments of phylogeny, we compute whole genome average nucleotide identity (ANI) for each pair of genomes using FastANI 1.32 [39]. Furthermore, whole-genome sequences of strains and related genomes were compared by using ANI-BLAST (ANIb) and ANI-MUMmer (ANIm) algorithms within the JSpeciesWS web service [40,41].

#### 2.9.4. Genome Annotation

Functional annotations based on clusters of orthologous groups of proteins (COGs) and protein families (Pfams 24.0) were performed with the webserver WebMGA (evalue = 0.001) [42] using the amino acid fast file obtained from RAST. Carbohydrate-active enzymes (CAZymes) were annotated based on HMMER searches (HMMER 3.0b) [43] against the dbCAN database release 9.0 [44,45].

### 2.10. Statistical Analysis

Statistical significance of the data was evaluated with the parametric t-test, Welch’s test (significance determined as 95%, *p* < 0.05), using the program R. The sample size, *n*, varied between 3 and 9.

## 3. Results

### 3.1. Comparison between Monospecific Cultures and Co-Cultures under Present CO_2_ Concentrations

Under present CO_2_ concentrations, *E. huxleyi* cell numbers (Figure 1) and growth rate (Figure 2) were not significantly affected by the presence of the bacteria tested. In contrast and under the same conditions, *I. abyssalis* abundances increased (~10^6^ CFU mL^−1^) significantly, but only after the longer incubation period (here called bloom phase) of coexistence with *E. huxleyi* (initial 4 days ~10^5^ CFU mL^−1^, Figure 3), while *Brachybacterium* sp. decreased to ~10^2^ CFU mL^−1^ during the same time frame. Accordingly, within the first nutrient replete (Appendix A) 4 days under present CO_2_ concentrations, estimated cellular element quotas for *E. huxleyi* (Figure 4 and Figure 5) and bacteria (Figure 6) did not vary significantly, apart from decreasing nitrogen (*p*_Brachybacterium_ = 0.03, Figure 4) and organic carbon quotas of *E. huxleyi* in coexistence with *Brachybacterium* sp. in relation to the axenic culture (*p*_Brachybacterium_ = 0.04, Figure 5). Although no difference in cellular PIC quotas was detected after the initial days as a result of coexistence with *I. abyssalis*. At the end of the incubation, coccolith distal shield width was significantly larger than other cultures (Table 1). Finally, at the end of the incubation period under present conditions, extracellular enzyme activities were low, but leucine aminopeptidase and β-glucosidase (only E + B) activities were higher in coexistence than in the single cultures (Table 2).

### 3.2. Comparison between Monospecific Cultures and Co-Cultures under High CO_2_ Treatments

Under high CO_2_ concentration, abundances (Figure 1) and growth rates (Figure 2) of axenic cultures of *E. huxleyi* were not significantly different from co-cultures (Figure 2). However, the two bacteria benefited from the presence of *E. huxleyi,* dramatically increasing in abundance under enhanced CO_2_ concentrations (from ~10^2^ CFU mL^−1^ to ~10^6^ CFU mL^−1^, Figure 3) after an initial lag phase (Figure 3). This longer coexistence and enhanced bacterial abundance corresponded to decreased growth rates of *E. huxleyi* in relation to the exponential phase (Figure 2). Within the initial 4 days (exponential period), cellular contents of *E. huxleyi* did not show significant differences in estimated particulate total nitrogen and carbon (Figure 4), decreasing only particulate inorganic carbon when *E. huxleyi* was co-cultured with *I. abyssalis* at enhanced CO_2_ concentrations (*p* = 0.057), resulting in the lowest average PIC/POC (Figure 5) and, at the end of the incubation, significant increase in the coccoliths’ distal shield width (*p* = 0.02), distal shield area (*p* = 0.02), central area length (*p* = 0.01), central area width (*p* = 0.04) and central area area (*p* = 0.01) in relation to the axenic culture (Table 1). At the same time, bacteria carbon quotas at enhanced CO_2_ concentrations were higher in co-existence with *E. huxleyi* (Figure 6).

### 3.3. Responses to Increasing CO_2_ Concentrations

*E. huxleyi* abundances increased similarly under either CO_2_ treatment in the first 4 days of incubation (exponential, Figure 1) and consequently with no significant difference in growth rate (Figure 2) or in the cellular quotas of axenic *E. huxleyi* treatments (Figure 4). However, the co-existence of *E. huxleyi* with *I. abyssalis* significantly increased the coccolithophore’s organic carbon (*p* = 0.002) and decreased inorganic carbon quotas (p=0.04), resulting in a PIC/POC decrease (*p* = 0.06) in relation to the axenic culture at increasing CO_2_ concentrations (Figure 5). At the end of the incubation, almost all measures of coccolith size, with the exception of coccolith central area width, were larger under present CO_2_ concentrations in the axenic *E. huxleyi*, while only coccolith central area length was significantly affected in the co-cultures (Table 1 and Table 3).

Although *E. huxleyi’s* cellular contents only varied significantly in the presence of *I. abyssalis*, estimated bacterial carbon quotas increased significantly (*p*_TPC_ = 0.05) with increasing CO_2_ concentrations in all conditions and for both bacteria (assuming that CO_2_-specific *E. huxleyi* cellular quotas did not vary with co-existence) (Figure 6). These changes in cellular quotas occurred despite virtually no variation in bacterial numbers within the shorter incubation (Figure 3).

Prolonging the incubation period (bloom phase), resulted in the absence of both bacteria cultured alone and of *Brachybacterium* sp. in all treatments except when cultured with *E. huxleyi* at enhanced CO_2_ concentrations. Specifically, the growth rate of the two bacteria in co-culture from day 4 to 13/14 was 0.5 for *Brachybacterium* sp. and 0.27 for *I. abyssalis* at increased CO_2_ concentrations and 0.38 in the latter under present CO_2_. The observed differences in growth rate might be related to the two bacteria distinct functional profiling (Figure 7 and Appendix A) according to the cluster of orthologous genes of proteins (COG) and protein family (Pfam)-based annotations (Figure 7 and Appendix A). However, genome information does not directly reflect expression, here we can only consider the potential of the two species. In the *Brachybacterium* sp. genome, a high number of CDSs were assigned to the COG class ‘Carbohydrate transport and metabolism’ (Class G, Figure 7), while in the *I. abyssalis* genome a high number of CDSs were assigned to the COG classes ‘Cell motility’ and ‘Signal transduction mechanisms’ (Classes N and T, respectively, Appendix A). Unlike *Brachybacterium* sp., the *I. abyssalis* genome possessed complete metabolic pathways for flagellum biosynthesis (flgBCDEFGHIKM and fliDEGHJKLMNOPLS) and motility (cheABRWZ and motABY). Moreover, only in the *I. abyssalis* genome were several protein domains identified found in TonB-dependent receptors (PF00593, PF03544, PF07660 and PF07715), involved in the transport of sideropheres, as well as vitamin B12, nickel complexes and carbohydrates (Figure 7, *Idiomarina abyssalis*). Finally, the observed differences showed concomitant higher β-glucosidase activities under enhanced CO_2_ concentrations and also leucine aminopeptidase in the cultures with *E. huxleyi* and *I. abyssalis* (Appendix A).

## 4. Discussion

### 4.1. Co-Existence in the Present Ocean

Phytoplankton–bacteria interactions play a crucial role in the balance between organic matter production, recycling and transport into the deep. How specific relationships influence biogeochemical cycling under present conditions is still not fully understood. Testing the relationships between a cosmopolitan coccolithophore and two bacteria with distinct functional profiles could hint at the complexity of the relationships occurring at a given timepoint. *Idiomarina* (γ-proteobacteria) and *Brachybacterium* (Actinobacteria) are frequently found in natural communities (e.g., [18,46]). γ-proteobacteria are often abundant in free living communities (e.g., [47]) and are well represented in nonaxenic cultures [15]), while *Brachybacterium* is relatively less abundant (Southern Atlantic, [48]). Contrary to Roseobacter, the presence of the bacteria tested in this study at abundances of ~10^5^ CFU mL^−1^ did not affect the average growth rate of *E. huxleyi* in relation to its axenic cultures under present CO_2_ concentrations. Additionally, no considerable changes were observed in most enzymatic activities and cellular quotas of the coccolithophore or the bacteria in the present ocean, with the exception of particulate nitrogen of *E. huxleyi* in co-culture with *Brachybacterium* sp. on day 4, potentially as a result of the decay of *Brachybacterium* sp. This is supported by the observation that *Brachybacterium* sp. is not able to grow alone under present CO_2_ conditions for a longer period of time, potentially due to being restricted to inorganic nutrients and insufficient dissolved organic matter from the medium and *E. huxleyi* exudates. *I. abyssalis* was isolated from a fresh (months) *E. huxleyi* culture (isolated from offshore Terceira in 2016), potentially indicating a long-term commensal relation between the two organisms. Although *E. huxleyi* did not appear to benefit, *I. abyssalis* increased its growth rate in the presence of the coccolithophore. This only occurred after the initial days of exponential growth, when *E. huxleyi* abundances were higher, increasing the probability of phytoplankton–bacteria encounter rate and the provision of carbon sources from the coccolithophore. This species-specific effect [12] agrees with the response found here and could have consequences for carbon recycling.

### 4.2. Co-Existence-Driven Changes on Organisms’ Response to the Future Ocean

Our results showed that the effects of rising CO_2_ concentrations on the physiology of *E. huxleyi* was affected by the presence of bacteria, which in turn were differently impacted by it. A significant decrease of the growth rate of *E. huxleyi* was found after the longer incubation (higher abundances) at higher CO_2_ concentration. More importantly, this previously described decrease (e.g., [6,49]) was augmented by the presence of bacteria. The growth rates of the tested bacteria were lower than those observed when the isolates were incubated in the rich Marine Broth (µ ~2), but similar to other isolates grown with 0.5 mL Zobell Medium in 1 l filtered seawater [50]. The growth of heterotrophic bacteria has been found to benefit from enhanced exudation of organic matter (e.g., carbohydrates) by phytoplankton and resulting aggregated polysaccharides in the form of TEP that occur under nutrient limitation at the end of phytoplankton blooms [10,51] or associated with the exposure to increased CO_2_ concentrations [9]. Enhanced exudation of *E. huxleyi* as a response to higher CO_2_/lower pH might, indeed, provide more substrate, but bacteria might also increase their organic carbon demand to function as was seen here and in previous studies [52]. Whether this is species-dependent is unclear, but it might be related to the observed dependence of survival of bacteria isolates on the stress response of *E. huxleyi*. Moreover, the two bacteria tested have distinct capacities for carbohydrate utilisation, potentially related to differences observed in their responses.

In the natural environment, senescent phytoplankton and aggregates are quickly colonised by bacteria (e.g., [53,54]) further degrading particulate organic matter into dissolved organic matter [55], which is then reutilised. However, bacteria also has the potential to stimulate aggregation of phytoplankton cells between species [56]. Bacterial production has been found to increase with increasing phytoplankton biomass (e.g., Chlorophyl *a*) and with concomitantly enhanced organic matter at increased CO_2_ concentrations [56,57]. While both bacteria in the present study appear to benefit from the coccolithophore’s exudations without negative effects on the growth of *E. huxleyi’s* (commensalism) under future CO_2_ concentrations, under present conditions, *Brachybacterium* sp. was unable to survive in association with *E. huxleyi*. Dissolved organic carbon is mostly comprised of total dissolved carbohydrates and protein components [57,58]. Hence, the difference between the response of the two bacteria might be related to the fact that *Brachybacterium* sp. was not able to be as responsive to the exuded dissolved organic matter as *I. abyssalis*. *I abyssalis* was seen to be a common associate of our isolates of *E. huxleyi,* but also sinking particles [18]. The reasoning could also be related to the presence of the complement of TONB-dependent transporters in the *I. abyssalis* genome, which is known to be involved with the uptake of several compounds such as large protein fragments [58,59]. In fact, *Idiomarina* was first isolated from seawater of the deep sea and characterised as having a relevant protein metabolism for its carbon source [60]. How much of the exudates stay in the water column as TEP will impact the retention versus remineralisation of particulate organic matter [61] and, therefore, the strength of the biological pump.

No differences between axenic and co-cultures were found for most analysed extracellular enzymatic activities after the first four days, potentially due to low bacteria abundances and reduced quorum sensing. In fact, previous studies found maximum Vmax of leu-aminopeptidase and β-glucosidase at the end of a mesocosm experiment with a natural community [56,57]. The strongest response was found for proteases (leucine aminopeptidase) in the co-cultures after the longer incubation, corroborating the difference in the TONB-dependent transporters discussed above. The activity of these hydrolytic enzymes would potentially enable an efficient utilisation of polymeric organic material exuded by *E. huxleyi*. After the longer incubation, monospecific cultures showed higher enzymatic activities under enhanced CO_2_ concentrations, in contrast to previous studies [62]. Higher *E. huxleyi* and bacteria abundances would increase the probability of cells encountering each other and also potentially stimulating quorum sensing signals and affecting their responses.

More broadly, contradictory results have been found for the effect of elevated CO_2_ concentrations/reduced pH on marine heterotrophic bacteria, from morphological modifications and a temporary inhibition of growth in the case of *Vibrio* sp. [62,63], to lower bacterial abundances due to higher viral lysis rates [63,64,65]. Studies considering CO_2_ levels relevant to projected concentrations (190 to 1050 µatm) showed varying bacteria total abundance during phytoplankton blooms with small direct effects of CO_2_ concentrations [64,66,67]. Moreover, γ-proteobacteria and a few rare taxa (increased abundance) were significantly affected by rising CO_2_, after nutrient addition [66,68] Despite differences in relative abundances, the activity of free-living bacteria has been found to remain unaltered in a mesocosm study [66,67] while a reduction of 0.5 pH units resulted in a twofold enhancement of the β-glucosidase and leu-aminopeptidase rates [56,57]. Many enzymatic processes involved in the bacterial utilisation of organic substrates were shown to be pH sensitive in previous studies. Indeed, the efficiency of leucine aminopeptidase has been seen to triple at the highest CO_2_ concentration (280 to 3000 µatm), both in terms of total activity and per cell [51]. In line with Yamada and Suzumura [69], protease activity rates expressed by *I. abyssalis* and *Brachybacterium* sp. were mostly undetectable during the incubation. The highest activity rates were observed at increased CO_2_ concentrations. β-glucosidase rates were similar per hour to previous results published by Grossart, [62], but showed different results concerning CO_2_ concentrations. Increased enzymatic hydrolysis [9,51,68,70] together with enhanced presence of gel particles that can be utilised as food and surface for growth [51] at enhanced CO_2_ concentrations stimulates carbon and nutrient cycling as organic matter is degraded.

## 5. Summary

The expected changes in a future ocean, such as carbonate chemistry, will affect different organisms differently. Although there is increasing knowledge on the responses of marine bacteria and phytoplankton to increasing CO_2_ concentrations, little is known about their interactions. Photoautotrophs play a crucial role in the carbon cycle while fixing carbon by photosynthesis. Much of the organic matter formed is then recycled by microbes in the surface ocean, the so-called microbial loop [71,72]. In fact, in the present ocean 50 to 96% of net marine primary production is remineralised in the surface ocean within the microbial loop by bacteria [71,73]. Enhanced CO_2_ concentrations is expected to affect planktonic communities, with consequences for the cycling of organic matter, namely by enhancing photosynthesis of several marine algae analysed (e.g., [73,74]) as well as the rate of organic carbon and TEP production [74,75], with important repercussions for organic matter production and export to the seafloor [76]. However, the biotic interactions are highly dependent on other environmental conditions, such as nutrient availability and organisms’ abundance, which affect physiological conditions of the phytoplankton cells.

The present study shows that the impact of increasing CO_2_ concentrations is affected by the interactions established between *E. huxleyi* and specific bacteria, which are key for the aggregate dynamics and cycling of organic matter [67]. The growth of both bacteria was enhanced when the bacteria were grown with the coccolithophore in the future ocean CO_2_ scenario, potentially increasing recycling of organic matter produced and, therefore, minimising the negative feedback to the atmosphere resulting from photosynthesis.

## Figures and Tables

**Figure 1 microorganisms-10-02461-f001:**
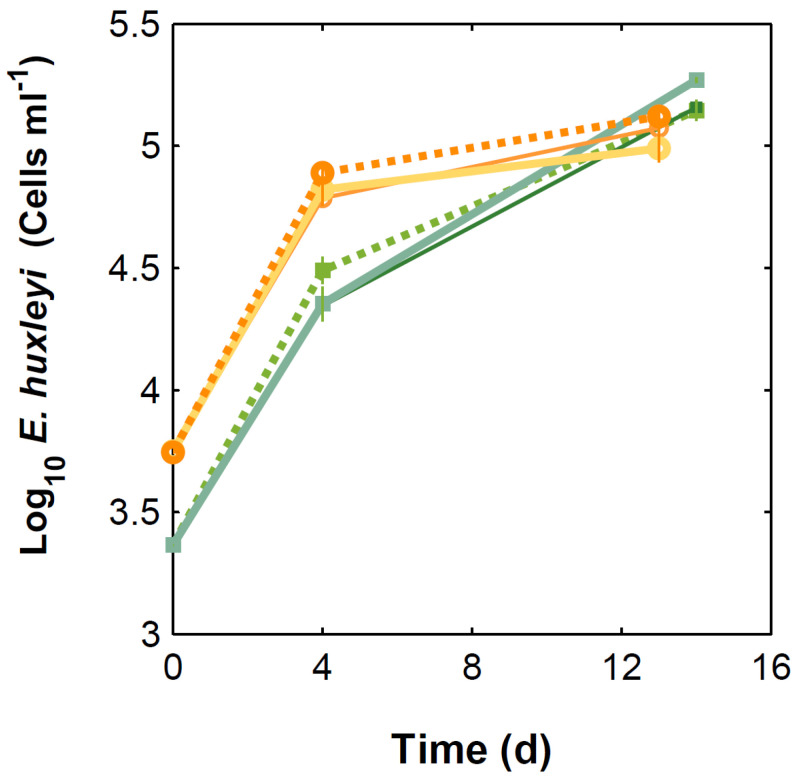
*E. huxleyi* abundances through time. Solid lines correspond to co-cultures and dashed lines to monocultures, under 475 (green) and 1056 µatm (orange). Thin solid lines correspond to *E. huxleyi* + *Brachybacterium* sp.; thicker solid lines to *E. huxleyi* + *I. abyssalis*.

**Figure 2 microorganisms-10-02461-f002:**
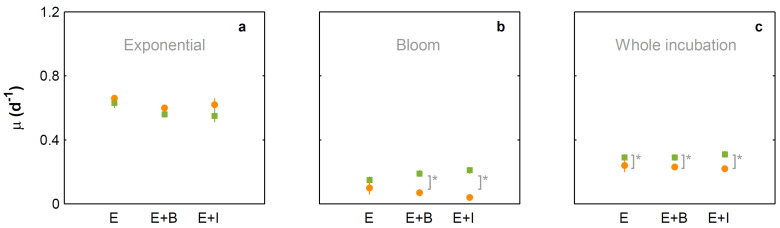
Growth rates based on cell counts of monocultures of *E. huxleyi* and *E. huxleyi* co-cultured with *Brachybacterium* sp. (E + B) and with *I. abyssalis* (E + I), under 475 (green) and 1056 µatm (orange). Cell division rates were calculated during the exponential (from 0 to 4 days, **a**), bloom phase (from 4 to 13/14 days, **b**) and whole incubation (**c**). * Denotes significant differences.

**Figure 3 microorganisms-10-02461-f003:**
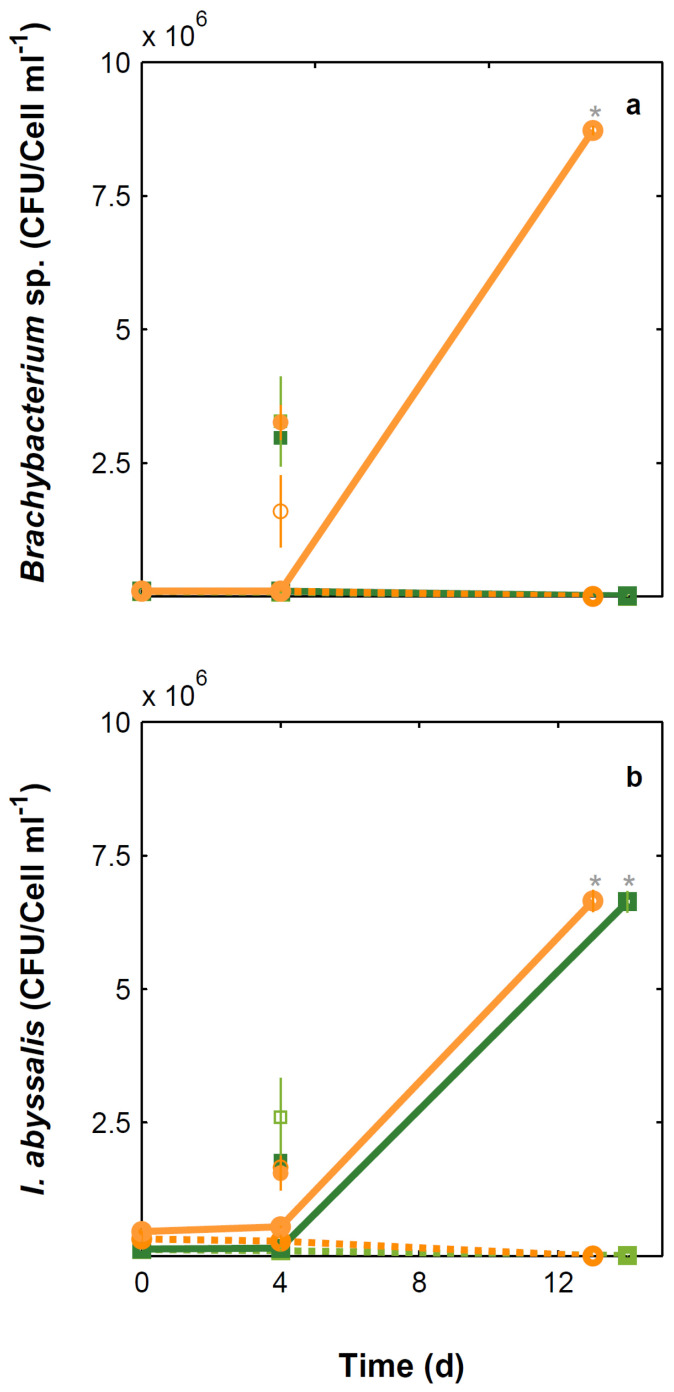
*Brachybacterium* sp. (**a**) and *I. abyssalis* (**b**) abundances through time. Lines refer to CFU data and isolated markers to counts obtained by means of flow cytometry. Solid lines correspond to co-cultures and dashed lines to monocultures of each bacteria isolate, under 475 (green) and 1056 µatm (orange). * Denotes significant differences.

**Figure 4 microorganisms-10-02461-f004:**
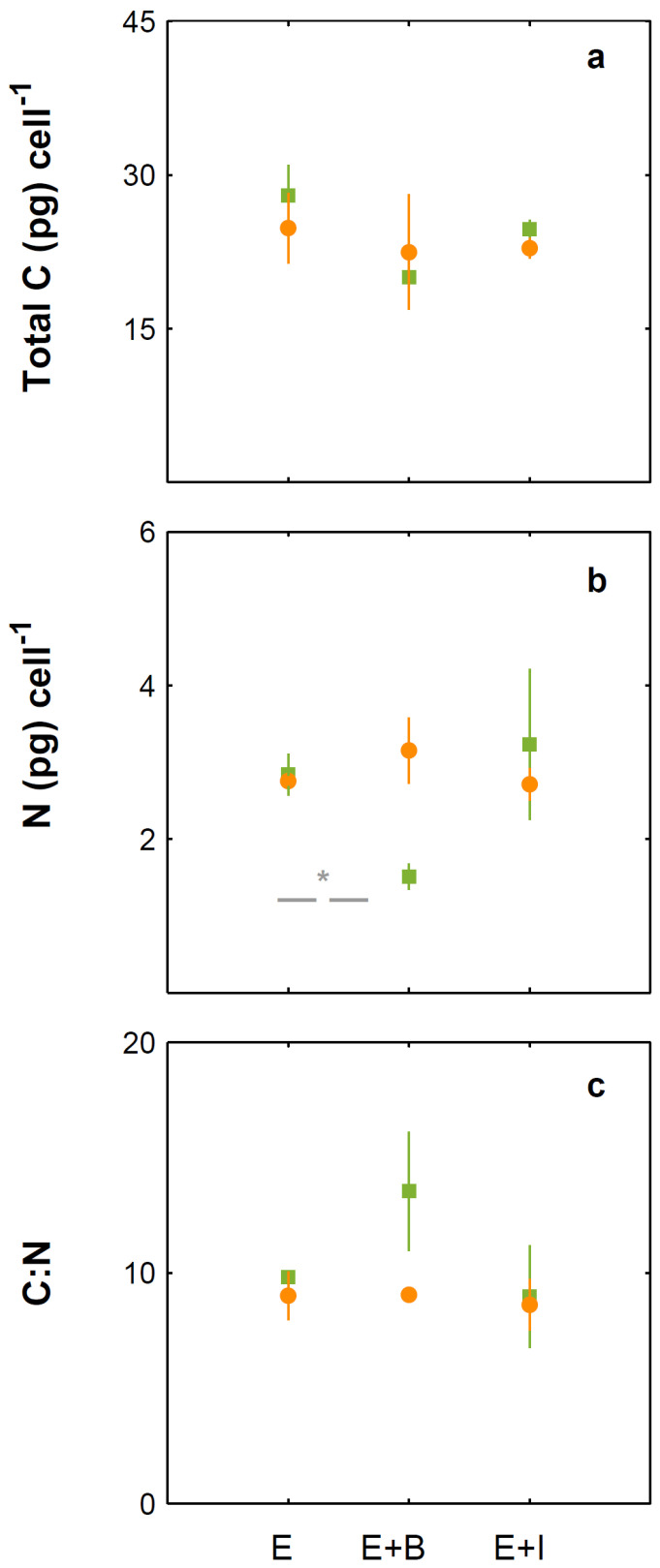
Cellular element quotas of *E. huxleyi* monocultures and *E. huxleyi* co-cultured with *Brachybacterium* sp. (E + B) and *I. abyssalis* (E + I), under 475 (green) and 1056 µatm (orange). Total particulate carbon (**a**), nitrogen (**b**) and carbon to nitrogen ratio (**c**). * Denotes significant differences.

**Figure 5 microorganisms-10-02461-f005:**
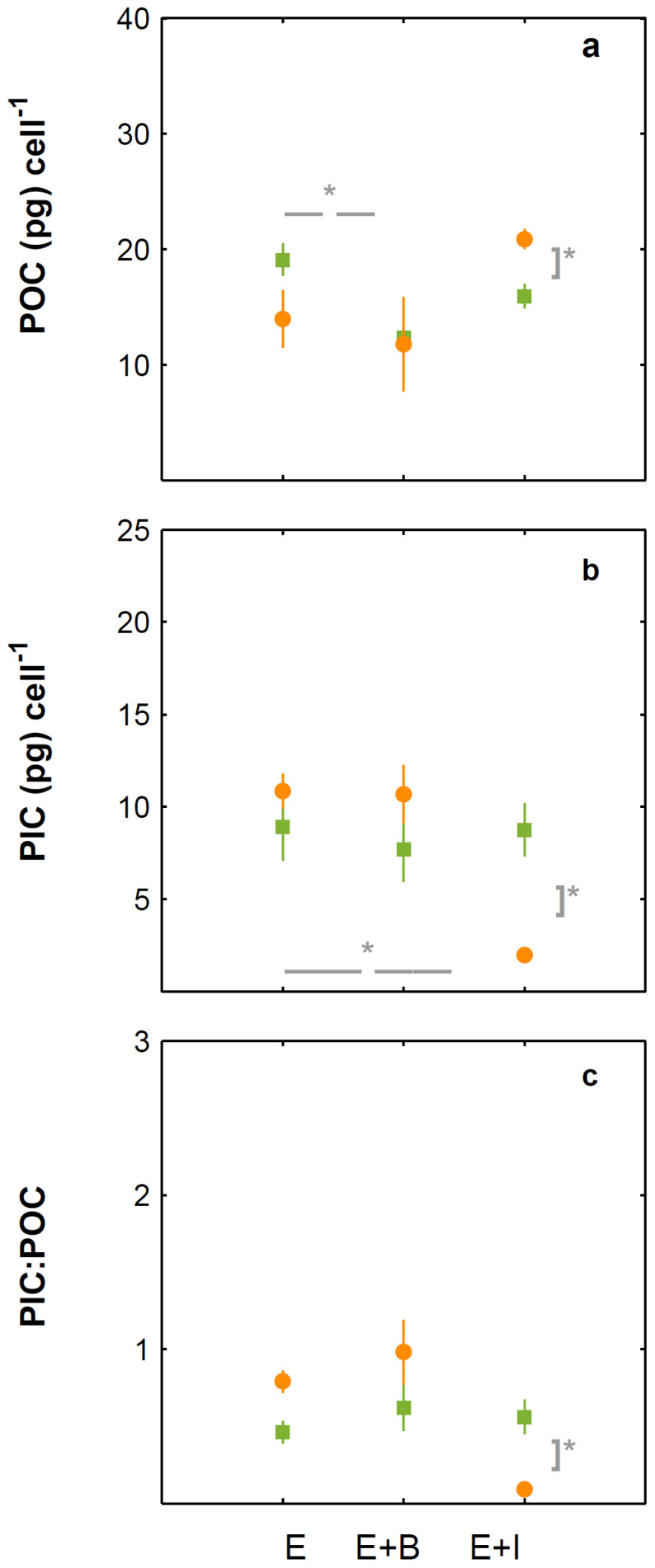
Cellular elemental quotas of *E. huxleyi* monocultures and *E. huxleyi* co-cultured with *Brachybacterium* sp. (E + B) and with *I. abyssalis* (E + I), under 475 (green) and 1056 µatm (orange). Particulate organic carbon (**a**), particulate inorganic carbon (**b**) and particulate inorganic to organic carbon ratio (**c**). * Denotes significant differences.

**Figure 6 microorganisms-10-02461-f006:**
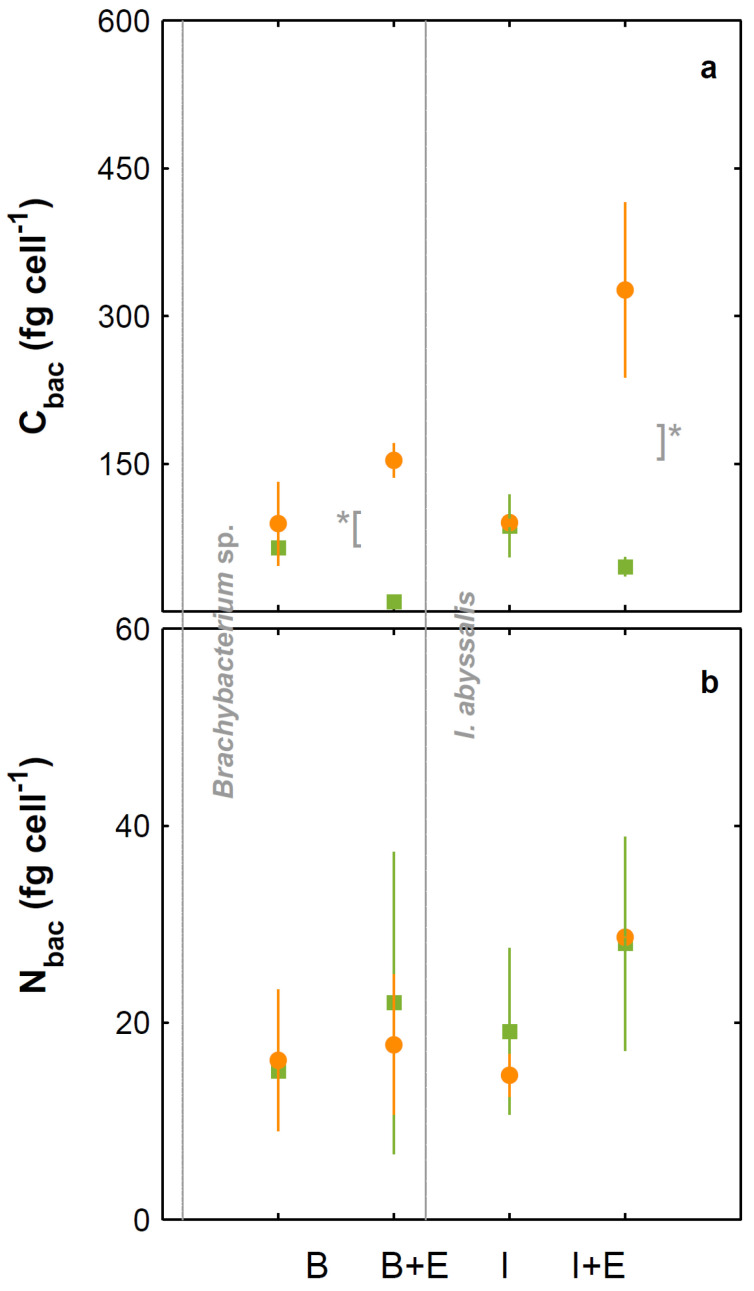
Cellular element quotas (based on total and organic particulate) of *Brachybacterium* sp. (B) and *I. abyssalis* (I) monocultures and those bacteria co-cultured with *E. huxleyi* (*Brachybacterium* sp., B + E; *I. abissalis*, I + E), under 475 (green) and 1056 µatm (orange). Total particulate carbon (**a**) and particulate nitrogen (**b**). * Denotes significant differences.

**Figure 7 microorganisms-10-02461-f007:**
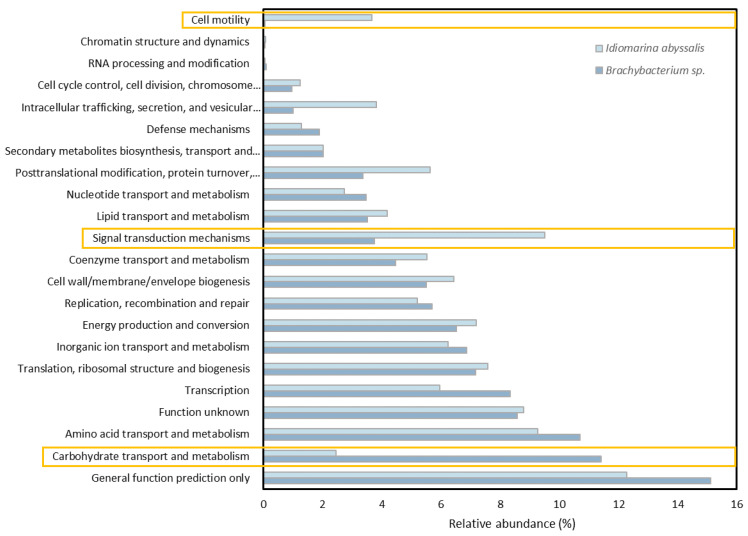
Percentage of genes assigned to specific functional activities of *Brachybacterium* sp. and *I. abyssalis*.

**Table 1 microorganisms-10-02461-t001:** Significance (*p* value) of *t*-test comparing coccolith measures and communities analysed. *E. huxleyi* (E) and *E. huxleyi* co-cultured with *Brachybacterium* sp. (E + B) and with *I. abyssalis* (E + I). Coccolith measures are distal shield length (DSL) and width (DSW), central area length (CAL) and width (CAW) and calculated distal shield area (DSA) and central area area (CAA).

	Present CO_2_						High CO_2_					
	DSL	DSW	DSA	CAL	CAW	CAA	DSL	DSW	DSA	CAL	CAW	CAA
E vs. E + B	0.26	0.9	0.6	0.6	0.3	0.3	0.3	0.7	0.6	0.6	0.6	0.5
E vs. E + I	0.4	0.02	0.07	0.07	0.6	0.6	0.07	0.02	0.02	0.01	0.04	0.01
E + B vs. E + I	0.73	0.02	0.2	0.2	0.1	0.1	0.9	0.44	0.6	0.6	0.5	0.5

**Table 2 microorganisms-10-02461-t002:** Enzymatic activities at the start (0) and after 4 and 13/14 days of incubation, under present and future CO_2_ concentrations.

		Leucine (µmol L^−1^ h^−1^)			Alkaline Phosphatase (µmol L^−1^ h^−1^)			α-Glucosidase (nmol L^−1^ h^−1^)			β-Glucosidase (nmol L^−1^ h^−1^)		
		0	4	13/14	0	4	13/14	0	4	13/14	0	4	13/14
p	*E. huxleyi*	0.02	0.03	0.14	0	0	0	0.79	0	0	0	0	0.42
		(+/−0.001)	(+/−0.009)	(+/−0.01)				(+/−0.39)					(+/−0.28)
f		0.05	0.12	0.42	0.3	0.18	0.43	0	0	0	0	0.57	7.21
		(+/−0.0006)	(+/−0.01)	(+/−0.02)	(+/−0.03)	(+/−0.07)	(+/−0.08)					(+/−0.57)	(+/−3.30)
p	*I. abyssalis*	7.89	0.01	0	157.64	0.18	0.14	0	0	0	0	0	0
		(+/−0.1)	(+/−0.002)		(+/−2.82)	(+/−0.08)	(+/−0.11)						
f		6.70	0.006	0	47.25	0.48	0.94	0	0	0	0	0	4.52
		(+/−0.08)	(+/−0.001)		(+/−0.63)	(+/−0.2)	(+/−0.03)						
p	*Brachybacterium* sp.	1.56	0	0	5.5	0	0.02	878.52	0	0	26.47	0	0
		(+/−0.02)			(+/−0.03)		(+/−0.01)	(+/−3.81)			(0.82)		
f		1.24	0	0	6.43	0.62	0.81	649.25	0	0	297.76	0	4.84
		(+/−0.01)			(+/−0.04)	(+/−0.18)	(+/−0.05)	(+/−7.12)			(+/−39.84)		(+/−2.27)
p	*E + I*	0	0.06	0.71	-	0.19	0.29	-	0	0	-	0	0
			(+/−0.005)	(+/−0.03)		(+/−0.03)	(+/−0.15)						
f		0	0.12	0.77		0.33	0.56		0	0		0	1.89
			(+/−0.02)	(+/−0.03)		(+/−0.11)	(+/−0.07)						(+/−1.95)
p	*E + B*	0	0.03	0.28	-	0	0	-	0	0	-	0	3.66
			(+/−0.003)	(+/−0.05)									(3.28)
f		0	0.12	0.51		0.3	0.38		0	0.02		0	4.24
			(+/−0.02)	(+/−0.03)		(+/−0.12)	(+/−0.10)			(+/−0.02)			(+/−2.73

Chitinase and Lipase below detection limits, Alkaline phosphatase is underestimated, since no buffer was added to improve fluorescence.

**Table 3 microorganisms-10-02461-t003:** Significance (*p* value) of *t*-test comparing coccolith measures and carbon dioxide conditions. *E. huxleyi* (E) and *E. huxleyi* co-cultured with *Brachybacterium* sp. (E + B) and with *I. abyssalis* (E + I).

	CO_2_ Effect					
	DSL	DSW	DSA	CAL	CAW	CAA
E	0.01	0.003	0.002	0.002	0.1	0.04
E + B	0.47	0.45	0.4	0.002	0.4	0.4
E + I	0.21	0.09	0.1	0.01	0.4	0.3

## Data Availability

The genome sequence of *Brachybacterium* sp. PhyBa_CO2_2 and *Idiomarina abyssalis* PhyBa_CO2_1 were deposited in GenBank database under accession number JAKJJW000000000 and JAOXLS000000000, respectively. 16S rRNA gene sequences of *Brachybacterium* sp. PhyBa_CO2_2 and *Idiomarina abyssalis* PhyBa_CO2_1 were deposited to GenBank under the accession numbers OM338106 and OP648247, respectively.

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
