# Peer review of "Emiliania huxleyi—Bacteria Interactions under Increasing CO2 Concentrations"

_microorganisms, 2022, doi:10.3390/microorganisms10122461_

Round 1

Reviewer 1 Report

This work analyzed the interaction between photoautotrophic organisms (Emiliana huxleyi) and heterotrophic bacteria (Idiomarina abyssalis and Brachybacterium sp.) to better understand the effects of increased CO2 concentration on the balance between export of organic matter to the deep ocean and recycling at the sea surface. The results obtained are a valuable contribution to the general knowledge of the impact of climate change on the microbial community and will contribute to a more accurate assessment of the potential impact of CO2 increase.

Table I: The table should be self-explanatory. Add meanings of abbreviations in the description

Figure 5: Add P(particular) to figures a and b; POC instead OC and PIC instead IC

Line 374: nitrogen carbon (Figure...nitrogen and carbon?

Reviewer 2 Report

This is an interesting study, highlighting the importance and need to study effects of anthropogenic stressors, such as increased CO2 concentrations on Emiliania-bacteria co-cultures and hence the effect on their interactions. The overall story is clear and well presented, however I have some questions concerning the experimental set-up and methods used that need clarification. Also, the visual support in form of figures should be improved, more details below.

In general, it would be very beneficial with an illustration of the experimental set-up.

Specific comments:

Experimental set-up:
It was not clear to me whether you measured the absence of bacteria in the E.huxleyi culture before you inoculated with the two different bacteria strains. In line 101/102 it states “the presence of bacteria in the E.huxleyi cultures was verified…” Independent, it should be clearly stated what the starting conditions were, and for this type of experiment it seems that it would be important that there are no other bacteria in the E.huxleyi cultures at the start before inoculation with the specific strains. Also, how many bacteria were added and what was the start concentration after addition?
Why was flow cytometry not used on day0 or day13/14 and more frequently?
I see the advantage of using CFU to determine viable cells, but for overall accuracy in quantification flow cytometry is of advantage. This also seems to be case when looking at your figure 3, where you detect cells at high abundances for all treatments when using flow cytometry at day 4 in both the co-cultures and the monocultures. Flow cytometry can also indicate the state of the bacteria (e.g. HNA vs LNA). Also live/dead staining of bacteria and microscopy analysis could have been done.
This brings me to another point that is important, you assume that the cells that you are counting are either Brachybacterium sp or I. abyssalis, but did you sequence some of the colonies that you counted at the different stages of the experiment or is it just assumed that they are those species because they were added?
Did you also test the abundance of bacteria in the E.huxleyi monocultures?
Regarding the decrease of Brachybacterium sp under 475 µatm, did you repeat the experiment and did you get the same result independently?

Figures:

In general some of the figures could be combined (figures 4 and 5) or illustrated in an easier way. You also have only three data points for the different abundance figures, so another type of graph with all three organisms combined could be an alternative (with two panels one for each CO2 condition).

Tables:
Table I: Here it is not clear what the abbreviations stand for.

Round 2

Reviewer 2 Report

Thank you for the thorough replies and corrections/improvements made to the manuscript. As written before, this is an interesting study and with most of my concerns being addressed this study has become much clearer. I think especially an illustration/scheme with the experimental setup will help, I could not find it in the documents that were provided, but I assume it will be included in the final version.
Overall, I think that figures could still be improved, but I understand the decision to keep the current style and appreciate that efforts have been made to try different ways.
Thank you for the clarifying my initial concerns and all the best with this publication.